

# Addressing the challenges of symbiont-mediated RNAi in aphids

Katherine M. Elston[1], Gerald P. Maeda[2], Julie Perreau[1,2] and Jeffrey E. Barrick[1]

[1] Department of Molecular Biosciences, The University of Texas, Austin, Texas, United States
[2] Department of Integrative Biology, The University of Texas, Austin, Texas, United States

## ABSTRACT

Because aphids are global agricultural pests and models for bacterial endosymbiosis, there is a need for reliable methods to study and control their gene function. However, current methods available for aphid gene knockout and knockdown of gene expression are often unreliable and time consuming. Techniques like CRISPR-Cas genome editing can take several months to achieve a single gene knockout because they rely on aphids going through a cycle of sexual reproduction, and aphids often lack strong, consistent levels of knockdown when fed or injected with molecules that induce an RNA interference (RNAi) response. In the hopes of addressing these challenges, we attempted to adapt a new method called symbiont-mediated RNAi (smRNAi) for use in aphids. smRNAi involves engineering a bacterial symbiont of the insect to continuously supply double-stranded RNA (dsRNA) inside the insect body. This approach has been successful in thrips, kissing bugs, and honeybees. We engineered the laboratory *Escherichia coli* strain HT115 and the native aphid symbiont *Serratia symbiotica* CWBI-2.3$^T$ to produce dsRNA inside the gut of the pea aphid (*Acyrthosiphon pisum*) targeting salivary effector protein (C002) or ecdysone receptor genes. For C002 assays, we also tested co-knockdown with an aphid nuclease (Nuc1) to reduce RNA degradation. However, we found that smRNAi was not a reliable method for aphid gene knockdown under our conditions. We were unable to consistently achieve the expected phenotypic changes with either target. However, we did see indications that elements of the RNAi pathway were modestly upregulated, and expression of some targeted genes appeared to be somewhat reduced in some trials. We conclude with a discussion of the possible avenues through which smRNAi, and aphid RNAi in general, could be improved in the future.

## INTRODUCTION

Aphids are tiny sap-sucking insects with outsized agricultural and biological relevance. They are worldwide pests responsible for the spread of devastating plant viruses (*Ng & Falk, 2006*; *Dedryver, Le Ralec & Fabre, 2010*; *Zapata et al., 2018*) and model species for understanding endosymbiosis (*Baumann, Moran & Baumann, 2006*; *Oliver et al., 2010*). They have a plethora of plastic traits, such as wing development under crowding and

Corresponding author
Jeffrey E. Barrick,
jbarrick@cm.utexas.edu

seasonal reproductive modes (*Grantham & Brisson, 2018*; *Parker & Brisson, 2019*). Despite these interesting phenotypes and the availability of genome sequences for several species, including *Acyrthosiphon pisum* (*Li et al., 2019*), there are limitations to the study of aphid genetics. Aphids primarily utilize a parthenogenetic reproductive mode in which they give birth to live offspring that are clones of their mothers. In the laboratory, it is difficult to induce aphid sexual reproduction, which is required for generating eggs and breeding offspring that are homozygous for new mutations. Additionally, aphids produce relatively few eggs (1–2 per ovariole) which then take about 100 days to hatch and typically only hatch successfully at low rates (*Trionnaire et al., 2008*). These limits make techniques like chemical mutagenesis and genome editing using RNA-guided nucleases (*e.g.*, CRISPR-Cas9) more challenging to perform (*Tagu et al., 2014*; *Le Trionnaire et al., 2019*). Because of this, researchers mainly rely on using RNA interference to knock down expression of aphid genes to study their functions.

RNA interference is not without its issues, however, and few researchers have used this technique to unravel complex genetic interactions (*Sun & Li, 2021*; *Wang et al., 2021*; *Ye et al., 2021*). RNAi is typically performed by either injecting or feeding aphids with an *in vitro* transcribed double-stranded RNA (dsRNA) with a sequence matching the aphid gene target. This dsRNA is processed by the aphid RNAi machinery, which results in knockdown of gene expression from the corresponding host mRNA. However, injection only delivers a limited dose of dsRNA, and nucleases in the aphid gut degrade dsRNA and reduce knockdown efficiency (*Christiaens, Swevers & Smagghe, 2014*; *Cao, Gatehouse & Fitches, 2018*; *Ghodke et al., 2019*). Researchers have overcome these issues by delivering higher, and continuous or repeated doses of dsRNA through feeding on transgenic plants or performing multiple injections (*Mao, Liu & Zeng, 2013*; *Ye et al., 2019a*; *Sun & Li, 2021*). However, these solutions have their own complications. Injections are traumatic for these tiny and fragile insects, and generating transgenic plants is time-consuming and untested for many combinations of aphid species and plant hosts.

Bacterial symbionts could serve as an alternative means of delivering dsRNA to insects. Bacteria are typically easier to genetically engineer than plants, and because symbionts are natively associated with insect hosts, they should robustly colonize and then continuously deliver dsRNA throughout the host's life. This methodology, called trans-kingdom RNAi in mammalian systems (*Xiang, Fruehauf & Li, 2006*; *Zhang et al., 2007*; *Nguyen & Fruehauf, 2009*) and symbiont-mediated RNAi (smRNAi) in insects, is effective at knocking down gene expression in thrips, kissing bugs, and honey bees (*Whitten et al., 2016*; *Leonard et al., 2020*). Our group recently developed methods for engineering the bacterium *Serratia symbiotica* CWBI-2.3$^{T}$ and showed that this culturable symbiont robustly colonizes the guts of multiple aphid species (*Elston et al., 2021*), making this system a prime candidate for testing smRNAi in aphids.

Here we describe our attempts to knock down *A. pisum* gene expression using symbiont-mediated RNAi. To inform our approach, we first performed an analysis of RNAi studies in aphids to select two targets, the C002 and ecdysone receptor (EcR) genes. Knockdown of these genes is expected to decrease aphid survival (*Mutti et al., 2006*), or increase the proportion of offspring that develop wings (*Vellichirammal et al., 2017*),

respectively. We then tested the efficacy of smRNAi using two different bacteria, *S. symbiotica* CWBI-2.3$^T$, which is naturally associated with aphids (*Sabri et al., 2011*; *Renoz et al., 2015*, *2019*; *Pons et al., 2019*; *Perreau et al., 2021*), and *Escherichia coli* HT115 DE3, which is an engineered laboratory strain that is used for RNAi knockdown in other invertebrates (*Dasgupta et al., 1998*; *Timmons, Court & Fire, 2001*). Despite some positive results in certain trials, we did not see reliable effects on aphid phenotypes or knockdown of mRNA levels with the methods tested. However, our negative results may prove valuable to others interested in pursuing symbiont-mediated RNAi in aphids and other insects and lead to a better understanding of the factors necessary for its success.

## MATERIALS AND METHODS

### Collection and analysis of aphid RNAi data

To gather the data for the comparison of aphid RNAi experiments, we read and manually parsed 86 scientific publications for information (Table S1). The aphid species tested, the target tested, the method of application of dsRNA, the length and concentration of the RNA, the type of RNA administered (*e.g.*, double-stranded RNA or small hairpin RNA), the level of gene knockdown, and the phenotypic effect of knockdown were recorded for each experiment in these publications. To calculate the total numbers of unique and repeat experimental designs, an experimental design was described as follows. Each experiment was first categorized by whether it delivered dsRNA by injection, feeding, or topical application. For injection experiments, a method was defined by the number of injections, the number of gene targets, and the concentration of dsRNA injected. For feeding experiments, a method was defined by the material fed on, the number of gene targets, and the concentration of dsRNA in the feeding solution. For topical application, a method was defined by the number of targets and the concentration of dsRNA applied to the aphid. All data was analyzed using R (Version 4.0.0; *R Core Team, 2020*).

### Maintaining bacterial and insect stocks

*Serratia symbiotica* CWBI-2.3$^T$ strains were grown using Trypticase Soy Broth (TSB) and Trypticase Soy Agar (TSA) at room temperature. *E. coli* HT115-DE3 strains were grown in LB at 30 °C. Media were supplemented where indicated with 0.3 mM diaminopimelic acid (DAP), 100 µg/mL carbenicillin (Carb), 20 µg/mL chloramphenicol (Cam), or 60 µg/mL spectinomycin (Spec). *Acyrthosiphon pisum* LSR1, PA1, 5A, and Tucson were acquired from long-term stocks maintained in the lab of Nancy Moran (University of Texas at Austin, Austin, TX, USA) and were reared on Broad Windsor *Vicia faba* plants (Mountain Valley Seed Company, South Salt Lake, UT, USA). Aphid cup cages were maintained in Percival I-36LLVL incubators (Perry, IA, USA) at 20 °C with a long (16L:8D) photoperiod. Aphids reared under these conditions have guts that are essentially sterile, and they are colonized in laboratory experiments only with the bacteria that are added to their diets (*Elston et al., 2021*).

## Preparation of dsRNA for injection and feeding experiments

*A. pisum* LSR1 RNA was extracted from a single aphid using the Zymo Quick RNA Miniprep Kit according to the manufacturer protocols (Zymo Research, Irvine, CA, USA). The RNA was then reverse transcribed using the High-Capacity cDNA Reverse Transcription Kit (Applied Biosystems, Waltham, MA, USA). Extended dsRNA regions were amplified from the cDNA template using their respective "Long" primers from Table S2. This product served as the template for the second PCR with the "T7" primers to amplify the target dsRNA fragment and add overhangs for T7 transcription. The T7 PCR template was used to prepare dsRNA with the HiScribe T7 Quick High Yield RNA Synthesis Kit according to the manufacturer protocols (New England BioLabs, Ipswich, MA, USA). For the injection experiments, dsRNA was purified with the Zymo Quick RNA Miniprep Kit. For the feeding experiments, LiCl precipitation was performed (*New England Biolabs, 2022*). The concentration of dsRNA was measured using a nanodrop, and the length of the transcripts was confirmed by running samples on a 1% agarose gel (*Nolan, Hands & Bustin, 2006*).

## Feeding aphids C002 dsRNA

For the feeding experiments, four genotypes of *A. pisum* (LSR1, PA1, 5A, and Tucson) were fed dsRNA in an artificial diet mix. Each diet mix contained 100 ng/μL dsRNA in a total of 100 μL Febvay diet + 1 μl yellow food dye (Gel Spice Company Inc., Bayonne, NJ, USA) (*Febvay, Delobel & Rahbé, 1988*; *Elston et al., 2021*). Sixty total 3–5-day old, second instar aphids were split into two feeding dishes per condition and fed for 18 h before being transferred to plants (20 aphids per plant, three plants per condition). Aphid mortality was monitored daily for a week and offspring were removed from the plants to prevent overcrowding.

## Engineering HT115 and CWBI-2.3$^T$ to express dsRNA

Construction of CWBI-2.3$^T$-GFP was described previously (*Elston et al., 2021*). HT115-GFP was built using pGRG36::PA1-GFP, a plasmid derived from pGRG-36 (*Phillips & Cooper, 2021*). To apply this system to HT115, we adapted methods from *McKenzie & Craig (2006)*. The donor *E. coli* strain, MFDpir, and the recipient HT115 strain were first grown overnight, then washed twice with 145 mM NaCl saline solution, mixed to a 1:2 ratio of donor:recipient, and 50 μL of the mix was spot-plated on LB + DAP agar. After a day of growth, the spot was scraped up and washed with saline as before. Dilutions of these bacteria were plated on LB + Carb agar and grown at 30 °C to select for transconjugants that had acquired the plasmid. Then, colonies were picked and re-grown on LB + Cam agar at 37 °C to select for transposition of the GFP cassette into the chromosome and loss of the temperature-sensitive plasmid. The resulting colonies were picked and regrown at 37 °C in liquid LB with Cam or Carb to confirm loss of the plasmid and successful integration.

The design of our dsRNA expression vector is based on a system for smRNAi in honey bees (*Leonard et al., 2020*; *Lariviere et al., 2022*). In brief, the plasmid has a low to medium copy RSF1010-based origin of replication, spectinomycin resistance, and a dsRNA target

sequence flanked by two identical inverted CP25 promoters. Each target for the RNAi experiments was first amplified from *A. pisum* LSR1 cDNA with "GGA" primers specified in Table S2, then cloned into the dsRNA site on the pBTK800 expression vector (*Lariviere et al., 2022*) with BsaI Golden Gate Assembly. The choice of amplified gene fragment was selected to replicate previous publications (*Mutti et al., 2006*; *Vellichirammal et al., 2017*; *Chung et al., 2018*). Assembled plasmids were conjugated or electroporated into CWBI-2.3$^T$-GFP and HT115-GFP for aphid feeding as described previously (*Elston et al., 2021*), except the dsNuc1-800 plasmid, which was tested in wild-type CWBI-2.3$^T$ and HT115 hosts. Plasmids used in this study are shown in Table S3.

## Symbiont-mediated RNAi targeting C002

Exponential phase cultures of HT115-GFP or CWBI-2.3$^T$-GFP with the indicated plasmids were grown in media with spectinomycin and then washed and resuspended in phosphate buffered saline (PBS) at an optical density of 1.0 at 600 nm (OD600). *A. pisum* LSR1 was reared to 5 days of age (late second instar) then fed on 98 μL of Febvay diet, plus 1 μL of resuspended bacterial cells and 1 μL of yellow food dye (*Elston et al., 2021*). Aphids were divided into 3–6 groups per condition and fed on the diet mix for 24 h, then transferred to the same number of respective plants. Their mortality was recorded daily for 1 week, and all dead aphids were crushed and plated on selective media as previously described (*Elston et al., 2021*) to confirm colonization. For the dsNuc1 co-inoculation experiments, 0.5 μL of the resuspended cell solution for each strain was added to the diet + food dye mixture.

## Injection of EcR dsRNA

To prepare for injection of EcR dsRNA, colonies of *A. pisum* LSR1 were reared at low density (5–7 aphids per plant) for at least two generations to decrease colony stress and the numbers of alate adults used in the experiments. Then, 45 10-day-old adult aphids were injected with ~0.1 μL of 2 μg/μL EcR or non-targeting E2-Crimson dsRNA. The injected aphids were allowed to recover overnight in petri dishes, then placed onto fava beans with five aphids per plant. Every 24 h the adults were transferred to fresh plants for a total of 3 days (*Vellichirammal et al., 2017*). The offspring were then reared to the 4th instar and scored for wing bud development.

## Symbiont-mediated RNAi targeting EcR

As described in the injection methods for EcR dsRNA, *A. pisum* LSR1 were first reared at a low density for at least two generations. CWBI-2.3$^T$-GFP with the dsEcR-800 or dsE2C-800 plasmid were grown and resuspended in PBS as described for the C002 experiments. Forty aphids per condition were fed overnight on a diet mix containing 98 μL of Febvay diet, 1 μL of blue food dye, and 1 μL of resuspended bacterial cells. After feeding, the aphids were transferred to plants and reared to adulthood (about 5 days post-feeding). At this point the aphids were screened for colonization with fluorescent CWBI-2.3$^T$ bacteria on a blue light transilluminator. Successfully colonized aphids were moved to fresh plants (four aphids per plant) and these groups were transferred to new plants every

24 h for 3 consecutive days. After 24 h on the final plants, these colonized adult aphids were removed to prevent the accumulation of additional offspring. Offspring produced by the colonized adults were reared to the 4th instar and scored for the development of wing buds.

### qPCR to assess gene knockdown in aphids

Bacteria were delivered to *A. pisum* LSR1 in artificial diet as described in the previous methods sections. For these experiments, five conditions were prepared: CWBI-2.3$^T$-GFP + BTK-NR, CWBI-2.3$^T$-GFP + dsE2C-800, CWBI-2.3$^T$-GFP + dsC002-800, CWBI-2.3$^T$-GFP + dsE2C-800 and CWBI-2.3$^T$ + dsNuc1-800, and CWBI-2.3$^T$-GFP + dsC002-800 and CWBI-2.3$^T$ + dsNuc1-800. Two sets of 20 aphids were treated to each condition; the aphids were then transferred to plants after 24 h of feeding. Five days after treatment, 6–10 aphids colonized with fluorescent bacteria from each condition were collected and crushed in DNA/RNA shield (Zymo Research, Irvine, CA, USA). Whole aphid RNA was purified using an RNA Clean & Concentrator kit (Zymo Research, Irvine, CA, USA) which includes an on-column DNA digestion step. Five hundred nanograms of each RNA sample were then reverse transcribed into cDNA using the Applied Biosystems High-Capacity cDNA Reverse Transcription Kit (Thermo Fisher, Waltham, MA, USA). Primers for qPCR were designed using IDT's Primerquest tool and are listed in Table S4. All qPCR reactions were prepared in triplicate in 384-well plates with SYBR Green reagents (Thermo Fisher, Waltham, MA, USA), and reactions were carried out using a ViiA-7 Real-Time PCR System (Thermo Fisher, Waltham, MA, USA).

Control qPCRs were run to determine the efficiency of each primer set and establish the most reliable reference genes for analysis. To determine primer efficiency, PCR products for each qPCR primer pair were synthesized, serially diluted, and run as the template to generate a standard curve. To choose the most reliable reference genes, we tested Beta-tubulin, GAPDH, NADH, Rpl32, and SDBH. We selected SDBH and Rpl32 to serve as references because they had the lowest variance between conditions. We calculated the gene expression ratio for each aphid target relative to the geometric mean of both reference genes by the Pfaffl method (*Pfaffl, 2001*). The amount of dsRNA production was determined by running the cDNA samples along with a standard curve for the gene of interest. The Cq values returned for the standard curve were then used to calculate the absolute copy number for each condition (*Bustin, 2000*).

## RESULTS

### Analysis of aphid RNAi experiments to select targets for knockdown

We wanted to select proven targets of RNA interference for testing our symbiont-mediated RNAi delivery system in aphids. To identify target genes expected to exhibit strong and reliable knockdown, we compiled a dataset of aphid RNAi results from 86 publications. We tabulated how RNA was administered, the effectiveness of gene knockdown, and other information for a total of 656 experiments (Table S1). By looking at the trends, we can get a sense of what is known about the reproducibility of different RNAi experimental designs in aphids (Fig. 1).
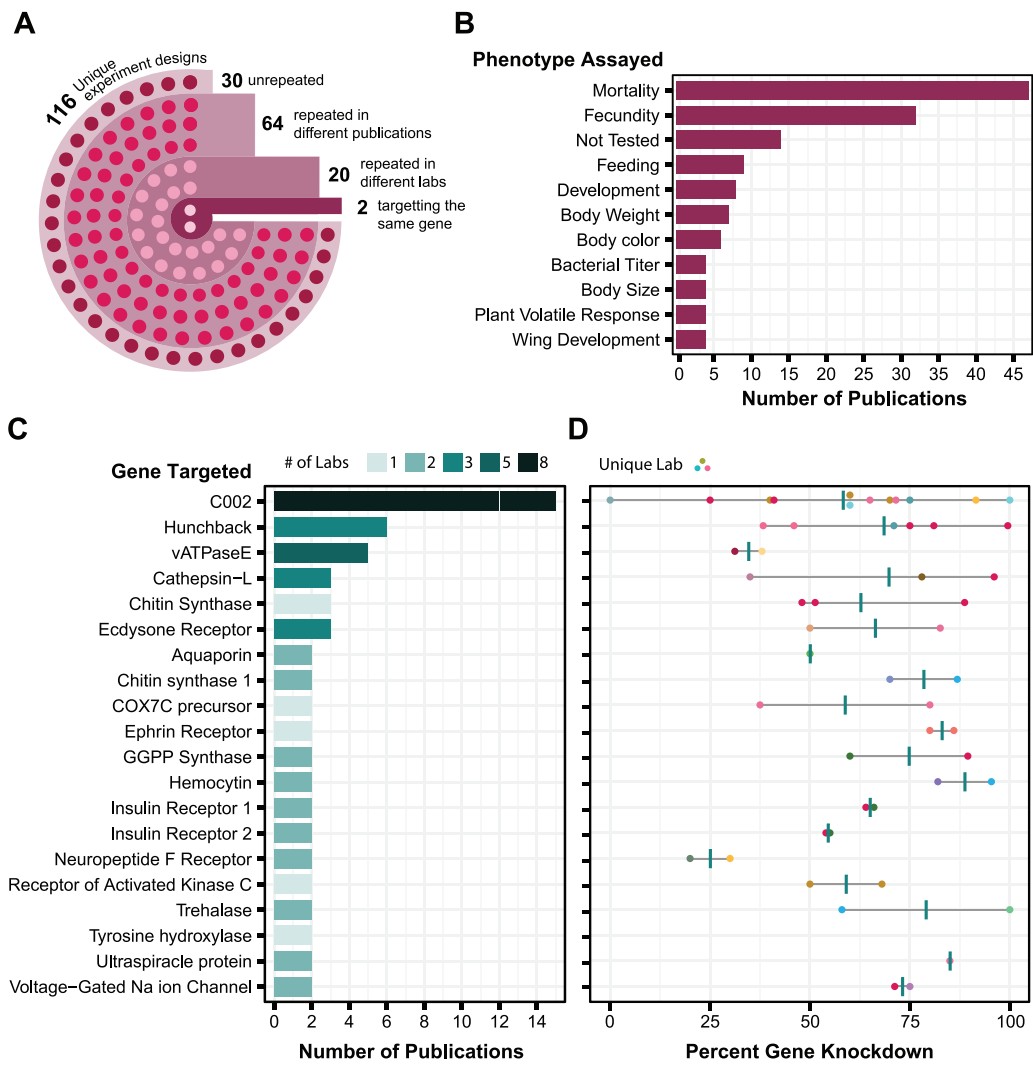

**Figure 1** **Analysis of aphid RNAi experiments.** Each plot is derived from data in Table S1 compiled from various publications. (A) RNAi experimental design. Each circle in the image reflects the number of experimental designs in each category. An experimental design is broadly described based on whether it is an injection, feeding, or topical technique. To further define an experimental design as unique, it is then categorized based on (where applicable) dsRNA concentration, number of targets, number of injections, and feeding source. (B) Phenotypes assayed in aphid RNAi experiments. The top 11 phenotypic outcomes of gene knockdown are plotted against the number of publications in which each was assayed. (C) Genes targeted in multiple aphid RNAi publications. Each target is plotted in relation to the number of papers it appears in and the number of labs that have assayed that target. Only targets that have been tested at least twice are shown. Shading is used to indicate the number of different labs that tested each target gene. (D) Corresponding levels of gene expression knockdown achieved in the experiments in C. Each point is colored based on the lab that performed the experiment and shows the maximum effect reported in each publication. Vertical bars are the average of these values.

The main challenge in using this information to judge reliability is that there are few examples of the same gene being targeted using the same RNAi procedure, and even fewer cases in which experimental conditions have been replicated by different labs. Out of the 116 different experimental designs used for RNA delivery, 86 have been replicated in
different papers, but only 22 of these have been repeated by different labs, and only two of these targeted the same gene (Fig. 1A). Most experiments with aphid RNAi are focused on aphid population management. Aphid mortality and reproduction are the phenotypes that are assayed most often (Fig. 1B). Experimental replication is also relatively low at the level of the genes targeted: out of 168 different gene targets, only 20 (12%) have appeared in more than one paper (Fig. 1C). The reported level of knockdown achieved *via* RNAi for these genes often varies greatly from study to study (Fig. 1D). Nevertheless, there are five targets (C002, hunchback, cathepsin-L, chitin synthase, and ecdysone receptor) for which the reported knockdown of gene expression by RNAi was more than 50%, on average, across RNAi experiments reported in at least three different publications.

The most highly tested target is the C002 gene, which was one of the first genes reported to be knocked down by RNAi in aphids (*Mutti et al., 2006*). This target has been tested by eight different laboratories and in multiple publications from some of these research groups (*Mutti et al., 2008*; *Pitino et al., 2011*; *Zhang et al., 2013*; *Pitino & Hogenhout, 2013*; *Christiaens, Swevers & Smagghe, 2014*; *Zhang et al., 2015*; *Coleman et al., 2015*; *Wang et al., 2015*; *Yan et al., 2016*; *Ghodke et al., 2019*; *Ye et al., 2019a*; *Sun et al., 2019*; *Niu et al., 2019*; *Jacques et al., 2020*) (Fig. 1C). Although there is high variability in the observed effectiveness of C002 knockdown (Fig. 1D), we decided that C002 would be the best benchmark for testing symbiont-mediated RNAi. Knockdown of C002 prevents aphid feeding and causes aphid mortality (*Mutti et al., 2008*), so we could examine both gene expression changes and this phenotype.

For a second target to test smRNAi we selected the ecdysone receptor (EcR). When EcR is knocked down it leads to an increase in the proportion of winged offspring birthed by the affected mother, which offers a phenotype to score that is different from the survival effect of C002 knockdown. RNAi results for EcR have been reported by two different labs in three papers using dsRNA delivery *via* feeding or injection (*Christiaens, Swevers & Smagghe, 2014*; *Yan et al., 2016*; *Vellichirammal et al., 2017*) (Fig. 1C), and the effectiveness of EcR knockdown appears to be similar to that observed for C002 (Fig. 1D).

## Effects of feeding dsRNA targeting C002

The effectiveness of RNAi varies in different aphid species and genotypes. We wanted to reproduce prior studies and select an aphid with the largest RNAi response (*Yoon et al., 2020*), so we assayed whether we could achieve knockdown of C002 in four different genotypes of *A. pisum*: LSR1, PA1, Tucson, and 5A. The C002 dsRNA sequence was designed to match the sequence in a prior study (*Mutti et al., 2006*), and the non-targeting dsRNA sequence, dsE2C, matches a 412 bp fragment of the E2-Crimson (E2C) fluorescent reporter gene not present in the aphid genome. We fed each aphid line on artificial diet supplemented with 100 ng/µL dsRNA for 24 h, then transferred them to plants and monitored their mortality for 6 days. Overall, mortality rates were low and none of the biotypes showed a significant increase in mortality when they were treated with C002 dsRNA in our experiment *vs* when they were treated with the non-targeting E2C dsRNA control (Fig. 2). Out of the four biotypes tested, we selected the two with the largest effects when testing whether there was a significant difference associated with dsRNA feeding—5A

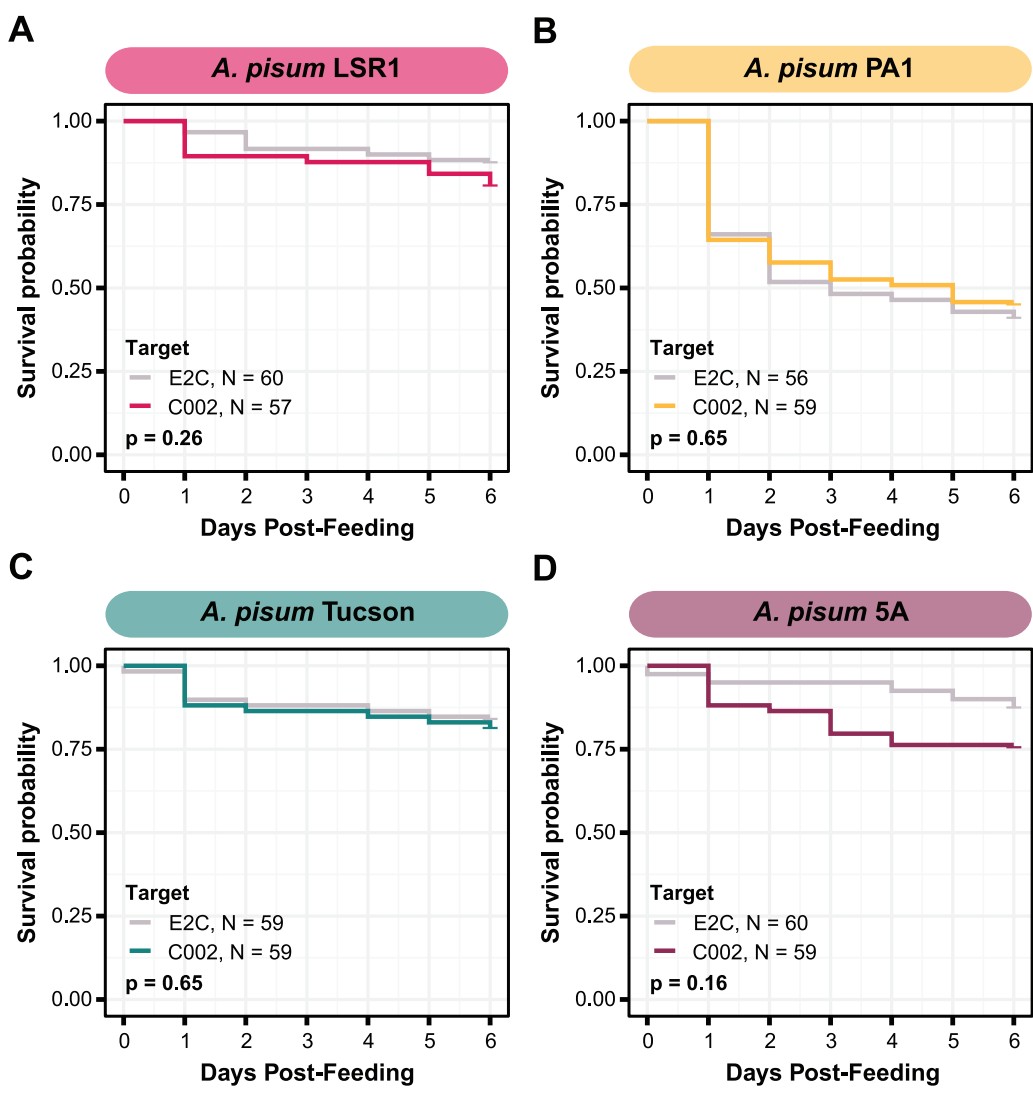

**Figure 2 Survival of *A. pisum* after feeding dsRNA targeting the C002 gene.** Each plot shows the survival probability of the aphids at each time point following feeding on C002 dsRNA compared to feeding on a non-targeting E2C dsRNA. Each panel shows the results for a different biotype of *A. pisum*, (A) LSR1, (B) PA1, (C) Tucson, and (D) 5A. *P*-values are calculated using two-sided log-rank tests.

(two-sided log-rank test, $N_{dsE2C} = 40$, $N_{dsC002} = 59$, $Z = 2.0$, $p = 0.16$) and LSR1 (two-sided log-rank test, $N_{dsE2C} = 60$, $N_{dsC002} = 57$, $Z = 1.3$, $p = 0.26$)—to see if smRNAi could achieve more effective results.

## Symbiont-mediated RNAi against C002 does not consistently lead to an increase in aphid mortality

In order to engineer our strains to produce dsRNA, we built plasmids based on the design used for smRNAi in honey bees (*Leonard et al., 2020*; *Lariviere et al., 2022*). The dsC002-800 and dsE2C-800 plasmids were constructed with the same C002 and E2C target sequences used in the feeding experiment by cloning these sequences into pBTK800, a

vector with two inward facing promoters that lead to expression of the insert as dsRNA. After building these constructs, we transformed them into versions of each of our bacteria that express green fluorescent protein (GFP) from a cassette integrated into their chromosomes, HT115-GFP and CWBI-2.3$^T$-GFP. We can easily identify aphids colonized with these strains because the fluorescent bacteria can be visualized inside the aphid gut on a blue light transilluminator (*Elston et al., 2021*).

We decided to first test smRNAi with HT115 *E. coli* because it has an RNase III mutation that increases accumulation of dsRNA. This mutation improves the effectiveness of RNAi in *C. elegans* (*Dasgupta et al., 1998*; *Timmons, Court & Fire, 2001*). We fed *A. pisum* on the dsRNA-expressing bacteria and monitored their mortality on fava beans for about 1 week (Fig. 3A). In our first trial with *A. pisum* LSR1, we observed a significant increase in aphid mortality in the C002 knockdown condition compared to the E2C knockdown control (two-sided log-rank test, $N_{dsE2C} = 22$, $N_{dsC002} = 17$, $Z = 9.8$, $p = 0.0018$) (Fig. 3B). However, when we repeated this trial with a larger number of aphids that were also divided into more enclosures (six instead of three per condition), we no longer observed a significant change (log-rank test, $N_{dsE2C} = 54$, $N_{dsC002} = 59$, $Z = 0.33$, $p = 0.56$) (Fig. 3C). We also did not observe a significant result in a smRNAi experiment in *A. pisum* 5A (two-sided log-rank test, $N_{dsE2C} = 38$, $N_{dsC002} = 38$, $Z = 1.8$, $p = 0.18$), though the average effect was in the expected direction (Fig. 3D).

Due to the high overall aphid mortality that resulted from colonizing aphids with *E. coli* HT115, we chose to test this knockdown with *S. symbiotica* CWBI-2.3$^T$ as well. Using the same experimental setup with three separate enclosures per condition, we delivered the dsRNA-expressing bacteria to *A. pisum* LSR1 and monitored their survival. Although the overall level of aphid mortality was lower with CWBI-2.3$^T$ (Fig. 3D) than with HT115 (Fig. 3B, Trial two) in the non-targeting controls (two-sided log-rank test, $N_{HT115} = 54$, $N_{CWBI} = 55$, $Z = 45$, $p < 0.0001$), we found the opposite result of what was expected for dsRNA expression. The non-targeting control condition with bacteria expressing E2C dsRNA had significantly higher mortality than the C002 condition (two-sided log-rank test, $N_{dsE2C} = 55$, $N_{dsC002} = 34$, $Z = 7.7$, $p = 0.0056$) (Fig. 3D). Overall, the relatively small effect size and substantial variation in these results indicates that we did not achieve a reliable increase in aphid mortality using HT115 or CWBI-2.3$^T$ as hosts for symbiont-mediated RNAi.

## Additional targeting of aphid Nuclease1 for knockdown does not increase the phenotypic effect of C002 symbiont-mediated RNAi

To attempt to improve the effectiveness of smRNAi we chose to co-target an aphid nuclease, Nuc1 along with C002. This approach has been shown to increase the potency of RNAi, most likely by decreasing dsRNA degradation in the aphid gut (*Chung et al., 2018*; *Yoon et al., 2020*). We first tested this technique with *E. coli* HT115. We co-inoculated aphids with two strains, one expressing dsC002-800, and one expressing dsNuc1-800, a construct built to target the same 328 bp of the Nuc1 gene as that were used in a prior study

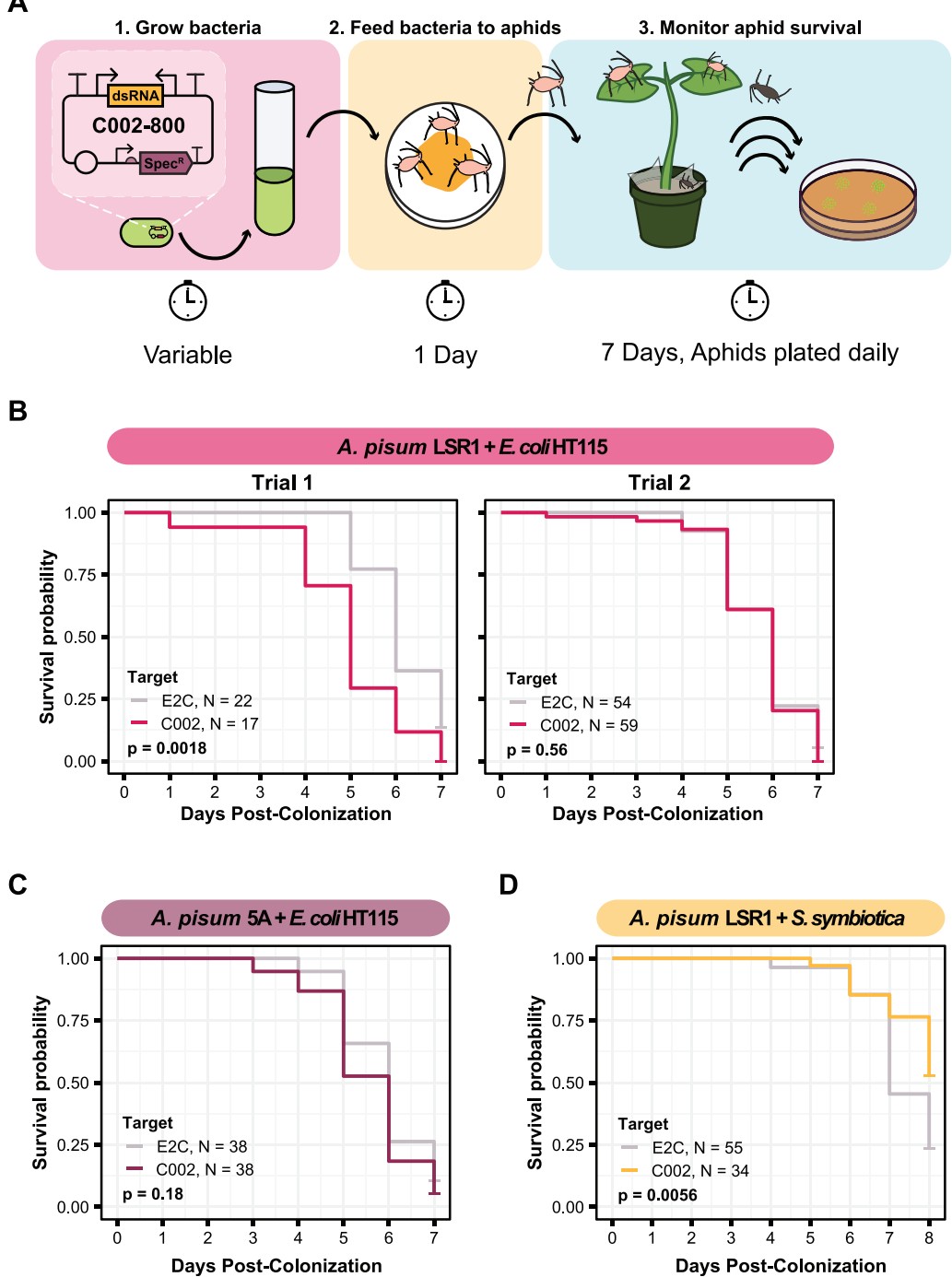

**Figure 3 Testing knockdown of C002 in aphids using smRNAi.** (A) Experimental procedure for the C002 assay. This figure shows the plasmid design for the system in the first panel along with the series of steps used to colonize the aphids with the bacteria and assay for a change in mortality. (B–D) Mortality curves for each of the smRNAi assays showing the survival probability of the aphids in the dsC002 condition compared to the dsE2C condition for each day after colonization with the bacteria. Sixty aphids were fed on the bacteria at the start of each experiment except for trial two of the *A. pisum* LSR1 + *E. coli* HT115 combination in which 120 aphids were fed. Only aphids that were colonized based on crushing and plating for bacterial growth were counted. *P* values were calculated using two-sided log-rank tests.

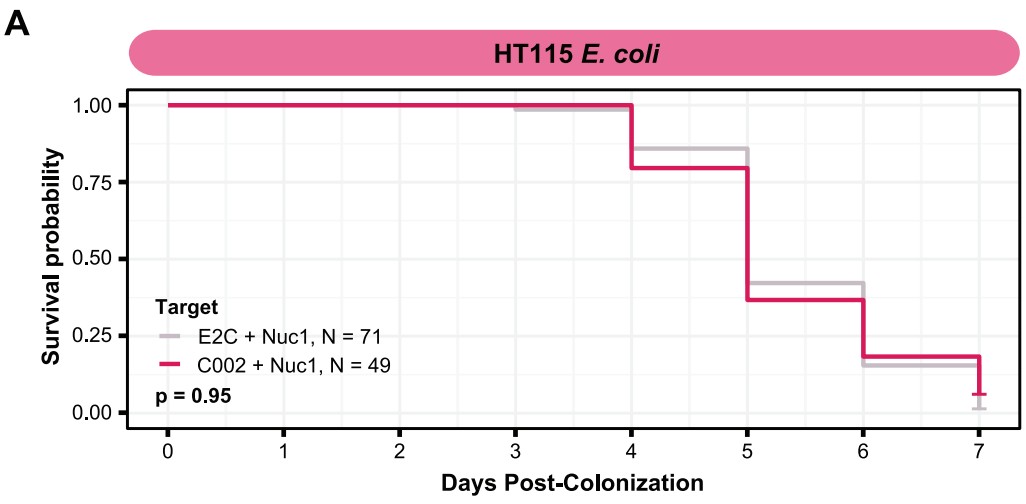

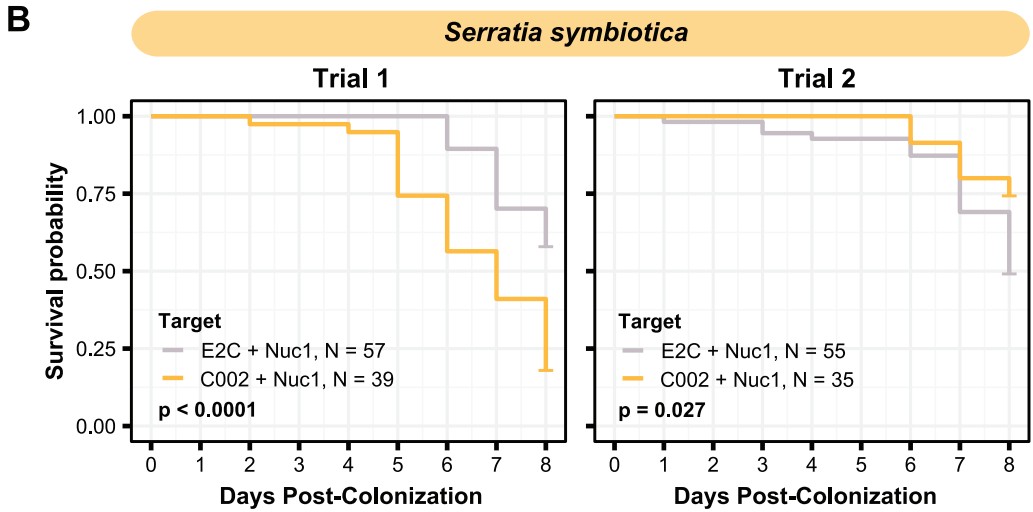

**Figure 4 Testing co-knockdown of C002 and Nuc1 using smRNAi.** (A) Mortality curve showing the survival of aphids treated with HT115-expressing dsRNA. (B) Mortality of aphid treated with *S. symbiotica* CWBI-2.3ᵀ as the delivery vector. Repeat trials are shown separately. Each panel compares the effect of the on-target C002 dsRNA combined with Nuc1 knockdown compared to the non-targeting E2C dsRNA with Nuc1 knockdown. Sixty aphids were fed on the bacteria at the start of (B) trial one, and 100 were fed for (A) and (B) trial two, but only aphids that were colonized based on crushing and plating for bacterial growth were counted. *P* values were calculated using two-sided log-rank tests.

(*Chung et al., 2018*), and monitored their mortality over 1 week. In our conditions, however, we did not see any improvement in RNAi effectiveness. Mortality in the HT115 double-target C002/Nuc1 condition was not significantly different than that observed for the E2C + Nuc1 control (two-sided log-rank test, $N_{dsE2C + dsNuc1} = 71$, $N_{dsC002 + dsNuc1} = 49$, $Z = 0.0033$, $p = 0.95$) (Fig. 4A).

To see if results would improve with the use of the native symbiont, we performed this same experiment using *S. symbiotica* CWBI-2.3ᵀ. The first attempt at this technique yielded encouraging results: mortality of the aphids in the C002/Nuc1 condition was

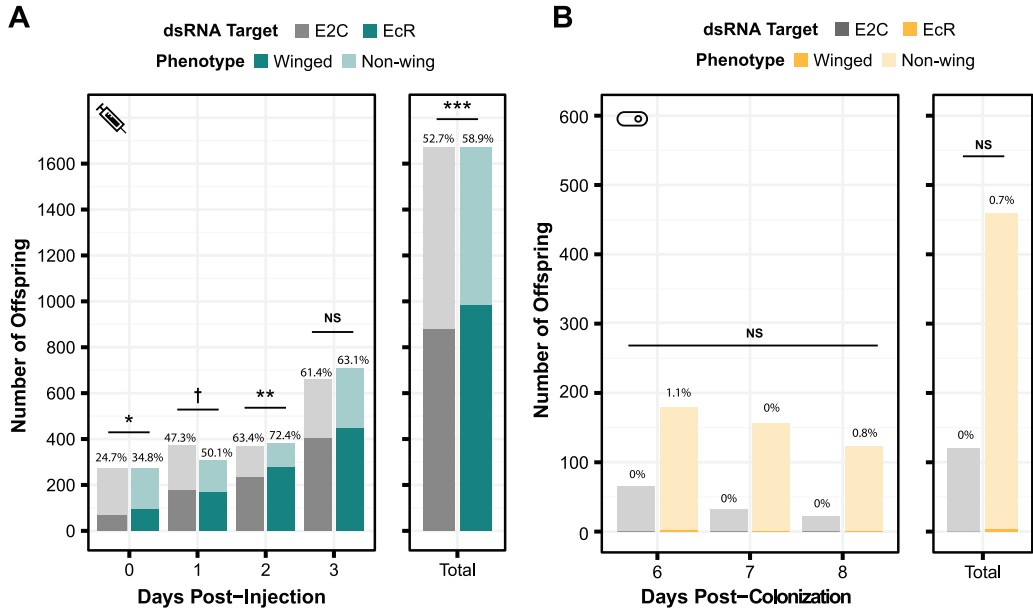

**Figure 5 Phenotypic results of testing EcR knockdown with two aphid RNAi methods.** (A) The percentage of winged to non-winged offspring birthed by mothers injected with E2C and EcR dsRNA are shown for each day post-injection along with the total for all 4 days. The offspring reflect the totals birthed from 45 mothers split onto nine separate plants. (B) The percentage of winged to non-winged offspring birthed by aphids fed on bacteria expressing dsE2C or dsEcR. A total of 40 mothers were fed bacteria per condition and then screened for successful colonization by examining them for GFP fluorescence. Colonized mothers were separated onto plants with four aphids per plant. *P* values were calculated using two-sided Fisher's exact tests. Significant differences are shown with symbols †*p* < 0.1, *\*p* < 0.05, \*\**p* < 0.01, \*\*\**p* < 0.001.

significantly increased compared to the control condition (two-sided log-rank test, $N_{dsE2C + dsNuc1} = 57$, $N_{dsC002 + dsNuc1} = 39$, $Z = 20$, $p < 0.0001$) (Fig. 4B). However, when we repeated this experiment under the same conditions, with increased numbers of aphids divided into five enclosures per condition instead of three, we saw the opposite of the expected result: aphids in the control condition died more than in the C002 condition (log-rank test, $N_{dsE2C + dsNuc1} = 55$, $N_{dsC002 + dsNuc1} = 35$, $Z = 4.9$, $p = 0.027$) (Fig. 4C). These phenotypic results demonstrate that even with the addition of Nuclease1 dsRNA, this technique is highly inconsistent.

## Knocking down EcR with smRNAi does not change numbers of winged offspring

To test the generality of our results, we performed smRNAi using *S. symbiotica* against a second gene target, EcR. Knockdown of this gene is expected to have a non-lethal phenotype. When the expression of this protein decreases, the proportion of winged (alate) offspring birthed by the treated mother increases. We first performed a control experiment where we injected aphids with EcR dsRNA to assess whether we could recapitulate the results of experiments by another group, using the same 466 bp sequence of EcR as their experiments (*Vellichirammal et al., 2017*). As shown in Fig. 5A, we observed a significant change in the number of winged offspring on days 0, 1, and 2 post-injection (two-sided

Fisher's exact tests, $N_{dsE2C} = 271$, $N_{dsEcR} = 273$, odds ratio = 1.6, $p = 0.011$, $N_{dsE2C} = 372$, $N_{dsEcR} = 307$, odds ratio = 1.3, $p = 0.090$, $N_{dsE2C} = 366$, $N_{dsEcR} = 380$, odds ratio = 1.5, $p = 0.0096$, respectively), though the effect did not persist through day 3 (two-sided Fisher's exact test, $N_{dsE2C} = 660$, $N_{dsEcR} = 710$, odds ratio = 1.1 $p = 0.54$). This mirrors the results of the original study, in which significant knockdown was only seen on the first day after injection (*Vellichirammal et al., 2017*).

Because we were able to achieve a significant phenotypic effect through dsRNA injection, we tested EcR knockdown with smRNAi. We designed our knockdown plasmid (dsEcR-800) to target the same 466 bp EcR sequence used in the injection test and transformed it into CWBI-2.3$^T$-GFP. Aphid stress plays a role in the production of winged offspring and seems to have been lower during this trial because the adults from both the on-target and non-targeting conditions had at most 1.1% winged offspring per day when we were monitoring them after colonization compared to the ~25–75% winged offspring that we observed in the injection experiment (Fig. 5B). This extremely low baseline, with almost no winged offspring at all in the smRNAi experiment, may have contributed to our inability to detect a significant effect from the EcR dsRNA treatment *vs* the non-targeting E2C dsRNA control (two-sided Fisher's exact test, $N_{dsE2C} = 121$, $N_{dsEcR} = 460$, odds ratio = 0, $p = 1.0$). We are unsure whether the added delay waiting for aphids to be fully colonized by *S. symbiotica*, an effect from *S. symbiotica* colonization itself, the lack of trauma from microinjection, or other, unknown environmental factors that varied in this trial gave rise to this inconsistency.

## qPCR to test for dsRNA expression and RNAi machinery upregulation

We performed qPCR to attempt to diagnose why the smRNAi system might be ineffective. To do so we set up a repeat of the C002/Nuc1 experiment with five conditions: aphids fed *S. symbiotica* containing an empty vector (pBTK-NR) or plasmids dsE2C, dsC002, dsE2C + dsNuc1, or dsC002 + dsNuc1. After 5 days we attempted to collect 10 colonized aphids from each condition for RNA extraction. However, there were few aphids with GFP+ guts in the dsC002, dsE2C + dsNuc1, and dsC002 + dsNuc1 conditions, so we only obtained seven, six, and three aphids from each, respectively. We did not further analyze the dsC002 + dsNuc1 condition due to its low sample size.

We first sought to confirm production of our dsRNA target sequences by *S. symbiotica* inside of the aphid. We cannot use qPCR to specifically verify production of C002 or Nuc1 dsRNA by engineered symbionts because the exact same sequences are also expressed as RNA by the aphid host, but this test is possible for the E2C non-targeting control. We used primers targeting E2C to estimate the copy number of dsRNA based on the amount of cDNA generated from 100 ng of total RNA. We were able to see clear expression of the dsRNA sequence in both the dsE2C and dsE2C + dsNuc1 conditions, although expression was lower in the paired condition (Fig. 6A). This difference could be due to unequal colonization by the dsE2C- and dsNuc1-producing bacteria that were co-inoculated in the paired condition.

For the treatments that included dsNuc1 or dsC002 we assayed the relative expression of the dsRNA construct compared to expression of the corresponding host gene using a set

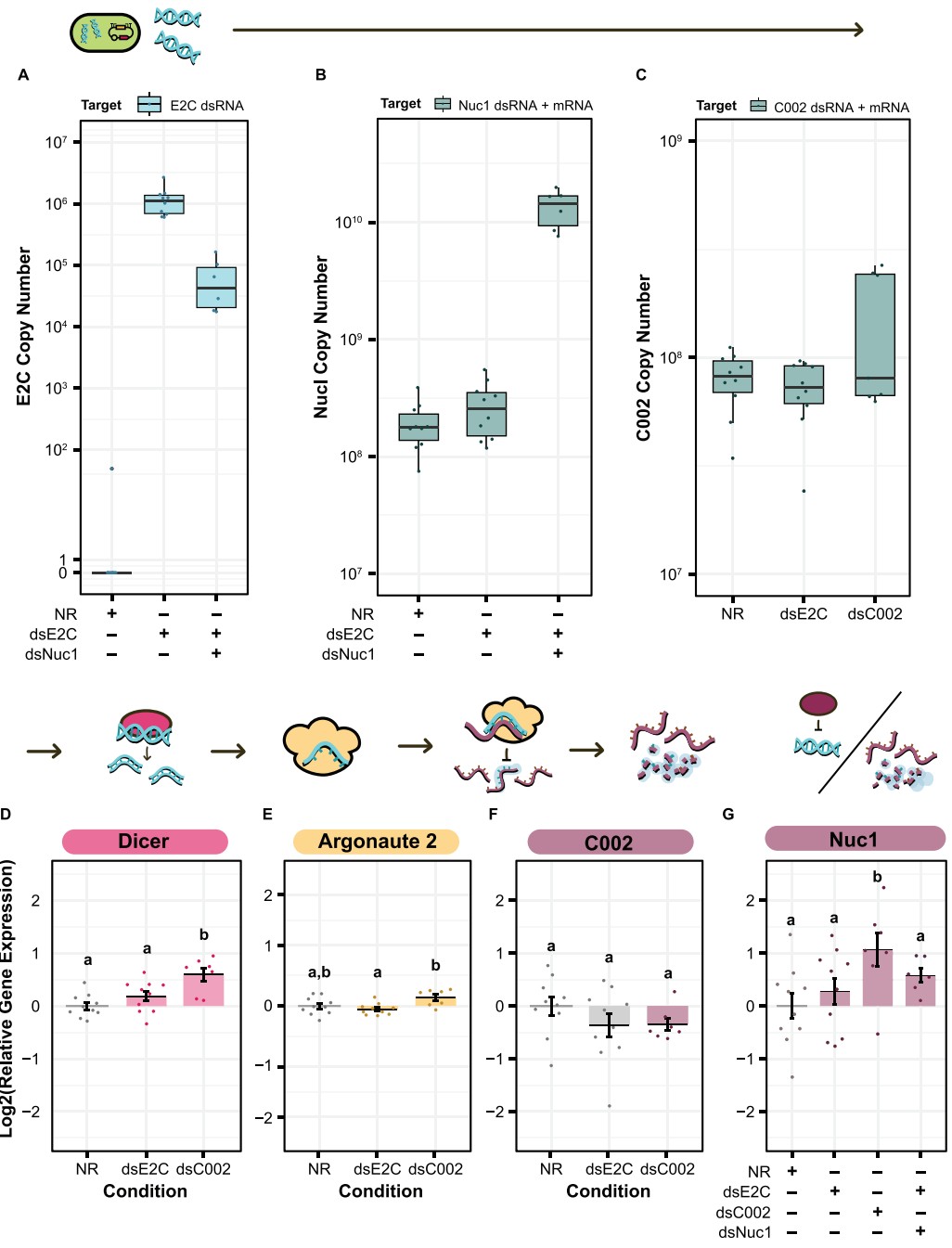

**Figure 6 qPCR for the level of dsRNA expression and RNAi machinery knockdown after smRNA against C002.** Above each panel is a simplified depiction of the RNAi process, from expression of dsRNA by the bacterium (green), to processing of dsRNA into siRNA by Dicer (pink), to recognition and knockdown of the matching host mRNA by Argonaute (yellow). (A–C) Copy number of (A) E2-Crimson, (B) Nuc1, and (C) C002 dsRNA sequence after treatment of the aphid with dsRNA expressing bacteria. Copy number is based on the amount of cDNA generated from 100 ng of total RNA. (B and C) For genes targeted by dsRNAs, copy number estimates include both the aphid mRNA for that gene plus any symbiont-produced dsRNA targeting the gene. (D–G) Relative levels of (D) Dicer, (E) Argonaute 2, (F) C002, and (G) Nuc1 expression calculated with the Pfaffl method. Letters signify groups based on a significant difference of $p < 0.05$. For all qPCR assays, a total of 6–10 aphids were measured for each condition. $P$ values were calculated using two-sided Mann–Whitney U tests.

of primers that targets a region found in both the dsRNA sequence and the aphid's mRNA for the gene (dsRNA + mRNA). The dsNuc1 construct causes an increase in the amount of the dsRNA + mRNA amplicon compared to the other control conditions, consistent with dsRNA expression from the symbiont (Fig. 6B). Expression of the dsC002 construct appears to be bimodally distributed (Fig. 6C). Three samples show an increased copy number for the dsRNA + mRNA amplicon, reflecting dsRNA production by the symbiont, while the other four samples have similar copy numbers to both control conditions which lack the dsC002 construct.

We next tested to see whether the RNAi pathway is upregulated by screening for changes in Dicer and Argonaute-2 (Ago-2). Dicer and Ago-2 were not significantly upregulated in the dsE2C condition (two-sided Mann–Whitney U tests, $N_{NR} = 10$, $N_{dsE2C} = 10$, $U = 31$, $r = 0.32$, $p = 0.17$, and $N_{NR} = 10$, $N_{dsE2C} = 10$, $U = 64$, $r = 0.24$, $p = 0.32$, respectively) (Figs. 6D and 6E). In the dsC002 condition, Dicer was significantly upregulated (two-sided Mann–Whitney U test, $N_{NR} = 10$, $N_{dsC002} = 7$, $U = 6$, $r = 0.69$, $p = 0.0031$) (Fig. 6D), but Ago2 is only marginally significantly upregulated compared to the control (two-sided Mann–Whitney U test, $N_{NR} = 10$, $N_{dsC002} = 7$, $U = 15$, $r = 0.47$, $p = 0.055$) (Fig. 6E). These results may indicate that the RNAi pathway has not been fully activated.

Finally, we tested whether smRNAi was effective at reducing C002 expression. For most of the dsC002 samples C002 expression is lower than in the no dsRNA (NR) treatment average. This effect is marginally significant (two-sided Mann–Whitney U test, $N_{NR} = 10$, $N_{dsC002} = 7$, $U = 51$, $r = 0.38$, $p = 0.13$) (Fig. 6F). Based on our qPCR data, it appears that we were able to produce enough dsRNA to activate Dicer and Ago-2 in the dsC002 condition which may be lowering expression of C002. This effect is still very slight, however, and the dsE2C condition also shows a knockdown trend, though it is more variable, so the reduction is not statistically significant (two-sided Mann–Whitney U tests, $N_{NR} = 10$, $N_{dsE2C} = 10$, $U = 64$, $r = 0.24$, $p = 0.32$) (Fig. 6F). We also measured our ability to knock down expression of Nuc1, but it is complicated by the general upregulation of this gene expected in the presence of dsRNA. This upregulation is significant in the dsC002 condition, but not in the dsE2C condition (two-sided Mann–Whitney U tests, $N_{NR} = 10$, $N_{dsC002} = 7$, $U = 11$, $r = 0.57$, $p = 0.019$, and $N_{NR} = 10$, $N_{dsE2C} = 10$, $U = 40$, $r = 0.17$, $p = 0.48$, respectively). For the dsE2C + dsNuc1 condition, we see marginally significant upregulation of Nuc1 (two-sided Mann–Whitney U test, $N_{NR} = 10$, $N_{dsE2C + dsNuc1} = 6$, $U = 14$, $r = 0.43$, $p = 0.093$), and the upregulation is higher than dsE2C alone. These results indicate that it may be possible to achieve knockdown of at least some aphid genes using smRNAi, but additional experiments are needed to understand where the system needs further optimization.

## DISCUSSION

In the configurations that we have tested, symbiont-mediated RNAi is not a reliable tool for gene knockdown in aphids. For both gene targets that we assayed we achieved inconsistent effects on aphid phenotypes. In the C002 trials, we saw the expected reduction in aphid survival in one trial but not in others. When testing the ecdysone receptor (EcR)

we observed so much variation in our baseline phenotypic outcomes that we could not draw reliable conclusions. Nevertheless, the results of our qPCR experiments show that dsRNA is being expressed by bacteria within the aphid body, that there is upregulation of genes associated with the host RNAi pathway, and some evidence suggesting that mRNA levels of the C002 gene are reduced when it is targeted.

We were careful to attempt to replicate our positive results, including additional preliminary experiments with fewer aphids that are not described here. If we had ended with the first promising result or restricted our reporting to specific gene targets or trials, then we may have judged the success of both injected dsRNA and smRNAi very differently. Some of the limited effectiveness and unreliability that we observed may be specific to smRNAi, such as the choice of symbiont and whether it can stably produce and release enough dsRNA to trigger an effect. Other complications are likely common to many types of aphid RNAi experiments, such as degradation of dsRNA in the aphid body, judging phenotypes that are highly variable, and other practical difficulties with replicating animal experiments such as enclosure-to-enclosure variation in the health of the aphids and the plants on which they feed.

The most important factor to consider for implementing smRNAi is the choice of symbiont. In our work we demonstrated a benefit of using a native symbiont instead of a laboratory *E. coli* strain. HT115 *E. coli* complicated our mortality assay and led to an increase in mortality for all aphids, whereas using *S. symbiotica* CWBI-2.3$^T$, reduced this complication. It is likely that choosing a well-adapted, native symbiont will generally be a benefit for implementing smRNA. Foreign strains may be under general attack by the host's immune system, which may limit their ability to colonize effectively and produce enough dsRNA to trigger effective gene knockdown. Pathogens that negatively affect host health are likely to have non-specific effects on host gene expression and phenotypes. Although less pathogenic than *E. coli*, *S. symbiotica* CWBI-2.3$^T$ is not a perfect symbiont choice in this respect. It can still reduce aphid longevity and increase aphid mortality (*Elston et al., 2021*; *Perreau et al., 2021*). If, in the future, a more host-adapted strain of *S. symbiotica* were to be cultured and/or genetically engineered, such as one of the bacteriocyte-associated strains, this would likely be a better symbiont for future aphid smRNAi trials.

Symbiont choice is also expected to affect the production and release of dsRNA. In our experiments we used a strain of *S. symbiotica* that retains expression of RNase III which may be a limiting factor in how much dsRNA accumulates in these cells. Knockout of this gene was unnecessary for successful smRNAi in honey bees using *Snodgrassella alvi* (*Leonard et al., 2020*), but RNase III knockout was essential to the success of using *Rhodococcus rhodnii* as a platform for smRNAi in kissing bugs and thrips (*Whitten et al., 2016*). Deleting RNase III or mutating it to reduce its expression if a complete knockout is inviable, would be a logical next step to test to increase the effectiveness of our system.

The dsRNA that is produced must also be released or secreted by the symbiont for delivery to the host. How this occurs during smRNAi is not well understood, even in systems where the technique has been successful. It is possible that lysis of a portion of the colonizing symbiont population continually releases dsRNA that is taken up by insect cells

in the gut in the same way as dsRNA fed in the diet. Aphids, like other insects, possess a *sid1*-like gene with sequence homology to the Sid1 transmembrane protein involved in intercellular transport of dsRNA in *C. elegans* (*Xu & Han, 2008*; *Huvenne & Smagghe, 2010*). However, altering expression of these *sid1*-like genes does not affect dsRNA uptake or systemic RNAi in other insects (*Tomoyasu et al., 2008*; *Luo et al., 2012*; *Zhu & Palli, 2020*), so a different mechanism is likely involved in aphids. For example, small noncoding RNAs can be packaged in outer membrane vesicles (OMVs), which have been proposed to play a role in the export of dsRNA (*Ghosal et al., 2015*; *Goodfellow et al., 2019*) and facilitate interspecies communication (*Lee, 2019*). OMVs are produced by most Gram-negative bacteria, but their abundance, size, and composition are variable (*Pathirana & Kaparakis-Liaskos, 2016*). Symbionts may naturally vary in whether they have access to different dsRNA delivery mechanisms. If the relevant mechanisms and genes were known or assays were developed for certain activities, then different symbionts could be computationally or experimentally screened to identify the most promising chassis for smRNAi.

Another general concern with implementing smRNAi in a new system is localization of the symbiont. In most cases, one wants the dsRNA produced by the symbiont, or at least derivatives of the dsRNA that result from processing in host cells, to spread from the symbiont cells and lead to knockdown of the target gene throughout the rest of the host's body. Because *S. symbiotica* CWBI-2.3$^T$ seems to be confined to the aphid gut, this is potentially a concern for our system (*Elston et al., 2021*). With this in mind, testing a validated gut-specific dsRNA target like *cathepsin-L* (*Sapountzis et al., 2014*) could be a way to establish reliable proof-of-principle results and a foothold for further optimizing aphid smRNAi in the future. It has been reported that dsRNA fed to aphids can enter other tissues (*Zhang et al., 2013*), and aphids are predicted to have systemic RNAi (*Xu & Han, 2008*; *Bansal & Michel, 2013*), as exemplified by the success of RNAi feeding studies (Table S1). However, it would be valuable to go back a step and confirm local smRNAi knockdown before further testing for systemic effects.

To parse whether production or delivery of dsRNA is the primary obstacle for our smRNAi system, studies with different experimental techniques are needed. We were able to observe production of our dsRNA sequences inside the host with qPCR, but this technique does not differentiate amplification products derived from single- or double-stranded RNA molecules or what proportion of the dsRNA was released from the bacterial cells. Because we only performed qPCR on whole aphids rather than on dissections of specific tissues, we also do not know to what extent symbiont-produced dsRNA propagates outside of the gut. To answer these questions, it may be possible to localize dsRNA in the aphid body with a dsRNA-binding antibody (*Stollar & Stollar, 1970*; *Son, Liang & Lipton, 2015*). This technique is widely used to detect dsRNAs associated with viruses and should be adaptable to our system (*Weber et al., 2006*). Ideally, dsRNA labelling would allow us to determine that double-stranded RNA, not single-stranded RNA, is expressed and ascertain how it leaves *S. symbiotica* cells. It could also be used to visualize the spread of dsRNA in the aphid body.

When implementing smRNAi, there may be further complications related to colonizing a host with a genetically engineered microbe. The dsRNA-expression construct may be burdensome to the symbiont or toxic to the insect. If it is burdensome, mutant cells that inactivate or reduce dsRNA expression may take over during culture in the laboratory or during insect colonization (*Renda, Hammerling & Barrick, 2014*). We observed bimodality in the production of C002 dsRNA between different colonized aphids, which may indicate that this type of evolutionary failure mode is occurring in our experiments. When the aphid phenotype being tested causes mortality, there will be a bias against observing colonization with any symbiont inducing a lethal RNAi response, which would likely further exacerbate this problem. Lastly, an effective insect RNAi response upregulates other immune pathways related to controlling bacterial pathogens which would increase the stress on the strain (*Brutscher, Daughenbaugh & Flenniken, 2017*). We encountered challenges achieving reliably high levels of *S. symbiotica* colonization in several C002 trials, which may be related to one or more of these factors. Whether or not burden or toxicity are affecting our system, they are important factors to consider when designing a smRNAi experiment as they can limit overall colonization or favor colonization with mutated symbionts that do not express dsRNA.

Overall, one key to improving this technique, along with RNAi in insects at a broader level, is to improve the reproducibility of these studies. We, like many others, have introduced a new methodology for dsRNA delivery. However, for the success of this field, there needs to be more emphasis on repeating the methods already tested by other groups to understand what ingredients from disparate protocols are critical for success (*Niu et al., 2019*; *Ye et al., 2019b*; *Yoon et al., 2020*; *Qiao et al., 2021*). A key to this endeavor will be determining which gene targets should be used as positive controls and benchmarks. As of now, there is no established standard in the field to ensure that an aphid RNAi procedure works in the hands of a new researcher in a different lab.

We attempted to establish a positive control for our aphid RNAi experiments based on the most popular gene target. However, prior results reported for the C002 gene still showed wide variation (*Christiaens, Swevers & Smagghe, 2014*; *Ghodke et al., 2019*), and it is hard to say whether this target is a reliable positive control when different methodologies are used in each publication. Because of this, perhaps the most popular gene target is not the best candidate for a reliable positive control. A better candidate may be found in survey papers where a variety of targets were tested side-by-side using a consistent RNAi delivery methodology (*Zhang et al., 2013*; *Ghodke et al., 2019*; *Ye et al., 2019a*; *Sun & Li, 2021*). The variability in our baseline results, particularly for the experiment with the ecdysone receptor, suggests that it may also be beneficial to test more genes that are decoupled from aphid viability and stress responses. Genes that affect insect appearance or general behaviors may be more reliable choices (*Tzin et al., 2015*; *Fan et al., 2015*; *Wang et al., 2019*; *Ding et al., 2019*; *Zhang et al., 2020*).

## CONCLUSIONS

In conclusion, although we tried to develop and test a new symbiont-mediated methodology to improve RNAi knockdown in aphids, we experienced a few pitfalls that

have generally plagued this field. Our results with smRNAi indicate that there may be potential for this technique using *Serratia symbiotica* CWBI-2.3[T]. However, it is likely that this symbiont would need to be further genetically modified, to improve dsRNA production and delivery, for example. If a more aphid-adapted symbiont is engineered in the future, it may be a better platform for this technique. Overall, reproducibility is a major challenge in aphid RNAi studies. Care should be taken to establish standard methodologies and select robust positive controls before it can be reliably used for research and pest control.

## ACKNOWLEDGEMENTS

We thank Nancy A. Moran for providing resources and advice on aphid genetics techniques, Sean P. Leonard for his extensive advice and providing strains and plasmids, Simon D'Alton and Peng Geng for their help during early stages of the project, and Sarah B. Bialik for assistance with aphid rearing. We thank Tim F. Cooper for the gift of plasmid pGRG36::PA1-GFP.

### Funding

This work was supported by the Defense Advanced Research Projects Agency (HR0011-17-2-0052), the U.S. Army Research Office (W911NF-20-1-0195), the National Science Foundation (DEB-1551092 to Nancy A. Moran), and a UT Austin University Graduate Continuing Fellowship to Katherine M. Elston. The funders had no role in study design, data collection and analysis, decision to publish, or preparation of the manuscript.

### Grant Disclosures

The following grant information was disclosed by the authors:
Defense Advanced Research Projects Agency: HR0011-17-2-0052.
U.S. Army Research Office: W911NF-20-1-0195.
National Science Foundation: DEB-1551092.
UT Austin University Graduate Continuing Fellowship.

### Competing Interests

Jeffrey E. Barrick is a co-inventor on a patent (US 11,382,989) covering the use of engineered symbionts to improve bee health.

### Author Contributions

- Katherine M. Elston conceived and designed the experiments, performed the experiments, analyzed the data, prepared figures and/or tables, authored or reviewed drafts of the article, and approved the final draft.
- Gerald P. Maeda conceived and designed the experiments, performed the experiments, analyzed the data, authored or reviewed drafts of the article, and approved the final draft.
- Julie Perreau conceived and designed the experiments, performed the experiments, analyzed the data, authored or reviewed drafts of the article, and approved the final draft.

- Jeffrey E. Barrick conceived and designed the experiments, analyzed the data, prepared figures and/or tables, authored or reviewed drafts of the article, and approved the final draft.

## Data Availability

The source data for all figures is available in the Supplemental Files.

## Supplemental Information

Supplemental information for this article can be found online at http://dx.doi.org/10.7717/peerj.14961#supplemental-information.

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
