# Peer review of "Addressing the challenges of symbiont-mediated RNAi in aphids"

_PeerJ, doi:10.7717/peerj.14961_

## Round 0.1 · original submission · Major Revisions

The proposed study is well-planned, well-described, well-executed, and must be confirmed. The authors clearly identify some of their methods' shortcomings and, where appropriate, what alternative mechanisms may be at work in this system. The only minor suggestion is that the description of the genes and their phenotypic effects be included in the introduction because the genes are mentioned in the methods but their phenotypes are not described until the results.

Reviewer 1 ·

Basic reporting

The study outlined is well-planned, well-described, well-executed, and essential to report. The authors have presented the material in a concise and informative manner. They clearly note where some of the shortcomings of their methods may be and, where appropriate, what alternative mechanisms may be operating in this system.

My only very minor comment is that the description of the genes and their phenotypic effects should be mentioned in the introduction since the genes are named in the methods, but their phenotypes are not described until the results.

Experimental design

The experimental design is exhaustive. The authors have detailed, appropriate and extensive controls throughout their experiments. Their presentation of the literature review methods and the corresponding figure are well-described, thorough, and informative.

Validity of the findings

Their results are well-motivated and seem in line with what they found in their experiments. The authors do a very excellent job of listing the mitigating factors that may have influenced their results and providing alternative explanations for some of their results, particularly those that don't align with the standard expectations of these methods.

Reviewer 2 ·

Basic reporting

The manuscript investigated whether symbiont-mediated dsRNA expression leads to RNAi in aphids. The authors engineered aphid symbionts to target well-known target genes and conducted bioassays to observe the phenotypic effects of gene silencing. They found that the silencing of COO2 gene expression reduced aphid survival. However, the outcome was variable between trials or among different aphid genotypes. Overall, the experiments were well-designed and executed. The authors also discussed several aspects of symbiont-mediated RNAi that need to be improved for future research. I have a few suggestions/questions to improve the manuscript.

L67: Change “These solutions have their own complications, though.” to “However, these solutions have their own complications.”

L202-205: There were 5 conditions, not 6. Was one treatment not described in that section?

L470-472: In C. elegans, Sid1 and Sid2 function as intercellular transport of dsRNA and dsRNA uptake, respectively. No insect homologs have been found (Zhu and Palli 2020 and references there in, https://pubmed.ncbi.nlm.nih.gov/31610134/).
Please revise that sentence accordingly.

Experimental design

no comment

Validity of the findings

no comment

Additional comments

no comment

·

Basic reporting

See attachment

Experimental design

See attachment

Validity of the findings

See attachment

Additional comments

See attachment

---

## Round 0.2 · accepted · Accept

All of the reviewers' concerns have been addressed by the authors.